# Expanded quantum vortex liquid regimes in the electron nematic superconductors FeSe$_{1-x}$S$_x$ and FeSe$_{1-x}$Te$_x$

M. Čulo [1,2,10] ✉, S. Licciardello[1,10], K. Ishida [3], K. Mukasa[3], J. Ayres [4], J. Buhot [4], Y.-T. Hsu [1,9], S. Imajo [5], M. W. Qiu[3], M. Saito [3], Y. Uezono[6], T. Otsuka[6], T. Watanabe [6], K. Kindo[5], T. Shibauchi [3], S. Kasahara[7], Y. Matsuda [8] & N. E. Hussey[1,4] ✉

The quantum vortex liquid (QVL) is an intriguing state of type-II super-conductors in which intense quantum fluctuations of the superconducting (SC) order parameter destroy the Abrikosov lattice even at very low temperatures. Such a state has only rarely been observed, however, and remains poorly understood. One of the key questions is the precise origin of such intense quantum fluctuations and the role of nearby non-SC phases or quantum critical points in amplifying these effects. Here we report a high-field magnetotransport study of FeSe$_{1-x}$S$_x$ and FeSe$_{1-x}$Te$_x$ which show a broad QVL regime both within and beyond their respective electron nematic phases. A clear correlation is found between the extent of the QVL and the strength of the superconductivity. This comparative study enables us to identify the essential elements that promote the QVL regime in unconventional super-conductors and to demonstrate that the QVL regime itself is most extended wherever superconductivity is weakest.

Many unconventional superconductors are highly anisotropic (low-dimensional), strongly type II, and often possess a non-$s$-wave order parameter. Such a unique combination of properties can give rise to novel vortex dynamics and thermodynamics, the study of which has created a rich field of both fundamental and technological interest[1–4]. Type-II superconductors, by definition, possess both a lower critical field, $H_{c1}$, beyond which magnetic flux first enters the material, and an upper critical field, $H_{c2}$, beyond which the vortex cores overlap and the normal state is restored. The region between $H_{c1}$ and $H_{c2}$ is composed of vortices which, in a perfectly homogeneous system, are arranged in a periodic (Abrikosov) lattice. On applying an external current, the flux lines experience a Lorentz force $F_L$ that causes the whole vortex lattice to move, and that can be counteracted only by the friction force. In such an unpinned vortex solid (VS), dissipationless transport is impossible, i.e., there is a finite resistivity at all temperatures below the SC transition temperature $T_c$[1] (see Fig. 1a).

Defects introduce pinning forces $F_p$ into the system that can counteract $F_L$ without invoking the friction force and thus preserve dissipationless transport over an extended field $H$ and temperature $T$ range. Since the system is no longer perfectly homogeneous, the vortex lattice usually transforms into a vortex glass. For simplicity, we refer here to both the vortex lattice and the vortex glass as VS phases.

[1]High Field Magnet Laboratory (HFML-EMFL) and Institute for Molecules and Materials, Radboud University, Toernooiveld 7, 6525 ED Nijmegen, Netherlands. [2]Institut za fiziku, Bijenička cesta 46, HR-10000 Zagreb, Croatia. [3]Department of Advanced Materials Science, University of Tokyo, Kashiwa, Chiba 277-8561, Japan. [4]H. H. Wills Physics Laboratory, University of Bristol, Tyndall Avenue, Bristol BS8 1TL, UK. [5]Institute for Solid State Physics, University of Tokyo, Kashiwa, Chiba 277-8581, Japan. [6]Graduate School of Science and Technology, Hirosaki University, Hirosaki, Aomori 036-8561, Japan. [7]Research Institute for Inter-disciplinary Science, Okayama University, 3-1-1 Tsushimanaka, Kita-Ku, Okayama 700-8530, Japan. [8]Department of Physics, Kyoto University, Sakyo-Ku, Kyoto 606-8502, Japan. [9]Present address: Center for Theory and Computation, National Tsing Hua University, No. 101, Section. 2, Kuang-Fu Road, Hsinchu 30013, Taiwan. [10]These authors contributed equally: M. Čulo, S. Licciardello. ✉e-mail: mculo@ifs.hr; n.e.hussey@bristol.ac.uk

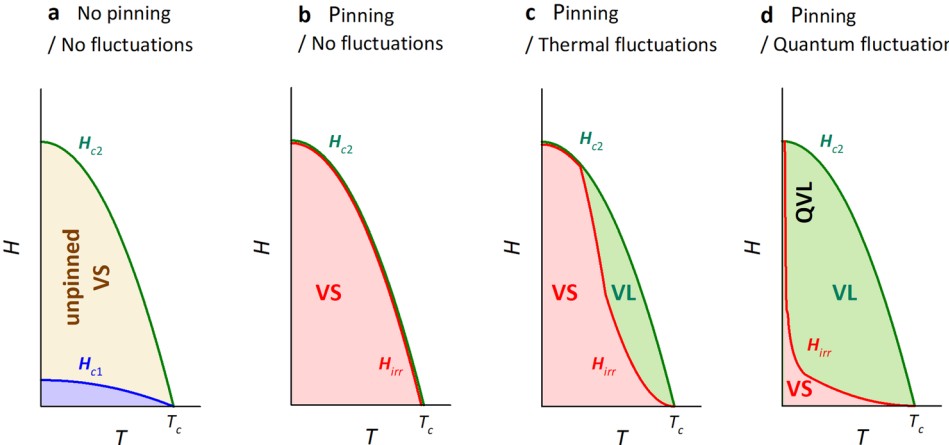

**Fig. 1 | Schematic $H-T$ phase diagrams of type-II superconductors. a** In a system with no pinning or fluctuations, the region between the Meissner phase (shaded blue below the $H_{c1}(T)$ line) and the normal phase beyond $H_{c2}(T)$ is occupied by the so-called unpinned vortex solid (VS) which has a finite resistivity for all $T$ below $T_c$. **b** In case of a finite pinning and negligible fluctuations, the $H_{irr}(T)$ curve lies close to $H_{c2}(T)$, and the region between $H_{c1}(T)$ and $H_{c2}(T)$ is a pinned VS with zero-resistivity for all $T < T_c$. (In panels **b**–**d**, the Meissner phase has been omitted for clarity.) **c** In the presence of both pinning and thermal fluctuations, the VS is destroyed, creating a wide vortex liquid (VL) regime and a marked separation between $H_{irr}(T)$ and $H_{c2}(T)$. At low-$T$, however, the two lines coincide. **d** The VS can also be destroyed by quantum fluctuations, giving rise to a quantum vortex liquid (QVL) phase. In this case, $H_{irr}(T)$ lies below $H_{c2}(T)$ even at $T \ll T_c$. Here it is worth mentioning that all phase diagrams refer to the limit $j \to 0$ where $H_{irr}$ line coincides with the melting line (see text).

In order to move the vortices and thus destroy the dissipationless transport, $F_L$ has to exceed $F_p$, which can occur either above a critical depinning current density $j_{dp}$ or above an irreversibility (or depinning) field $H_{irr}$[1]. (Here, we focus on the limit $j \to 0$ so that only $H_{irr}$ is the important field scale, and any complications with self-fields can be ignored.) In the case of strong pinning, often realized by artificial treatment of a superconductor, $F_L < F_p$ and, as a result, $H_{irr}(T) \sim H_{c2}(T)$. In the absence of fluctuations, the region between $H_{c1}$ and $H_{c2}$ is occupied by the pinned VS, which exhibits a zero-resistive state for all $T < T_c$ (see Fig. 1b). The situation becomes more complicated if the pinning is weak, since then in general $F_L$ can become larger than $F_p$ at some $H_{irr} < H_{c2}$ in which case the VS can become unpinned. Such complications, however, can be discarded in the limit $j \to 0$ we are interested in, where we can safely assume that $F_L$ is always smaller than $F_p$.

$H-T$ phase diagrams become much richer when we also take thermal fluctuations of the SC order parameter into account. These fluctuations can be understood as fluctuations in the position of the vortex lines, i.e., in the phase field of the SC order parameter. (Besides phase, there are also amplitude fluctuations of the SC order parameter not related to a vortex motion. Those fluctuations, however, are usually confined to a narrow region around $T_c$ and $H_{c2}$.) At the melting field $H_m(T)$, these thermal fluctuations become of order the vortex spacing, and the vortex lattice melts into a VL. In the VL phase, vortices move freely so that, on average, $F_p = 0$, $F_L > F_p$, and dissipationless transport is lost. In addition to $H_{c1}$ and $H_{c2}$, $H_m(T)$ is a fundamental quantity that shapes the $H-T$ phase diagram of type-II superconductors. It is worth mentioning here that $H_m(T)$ refers to a real solid-liquid transition, i.e., it is a thermodynamic quantity, while $H_{irr}(T)$ refers to the onset of a finite resistivity arising from either the solid-liquid transition or from a simple unpinning of the VS. In other words, $H_{irr}(T)$ does not necessarily point towards the VS-VL transition and is, in general, always $\leq H_m(T)$. In the limit $j \to 0$, $H_m(T) = H_{irr}(T)$, thus enabling the determination of $H_m(T)$ from dc transport measurements. A schematic representation of the $H-T$ phase diagram in the presence of thermal fluctuations is shown in Fig. 1c. As we can see, the thermal fluctuations destroy the VS across a large portion of the phase diagram at elevated temperatures.

If in addition to these thermal fluctuations, the system is also subject to quantum fluctuations of the SC order parameter, the VS can melt even at $T \ll T_c$, resulting in the formation of a quantum VL (QVL) (see Fig. 1d). While initial theoretical work explored the role of quantum fluctuations at finite temperatures[5], later work considered quantum melting of the vortex lattice even in the zero-temperature limit[6–8]. Experimental evidence for QVL formation, however, has only been reported for a small number of systems, including certain amorphous thin films[9–11], the organic conductor $\kappa$-(BEDT-TTF)$_2$Cu(NCS)$_2$[12–15], and low-doped high-$T_c$ cuprates[16], suggesting that conditions for its realization are rather stringent.

What is not evident from previous studies is whether the QVL and associated SC quantum fluctuations can be influenced by a nearby phase. Studies on amorphous films have shown that an increase in the normal state resistivity $\rho_n$ leads to an expansion of the QVL regime[17], but it is not clear whether a similar correlation exists in the more strongly correlated organic and cuprate superconductors. Many correlated metals also harbor a quantum critical point (QCP) somewhere in their $T$ vs. $g$ phase diagram, where $g$ is some non-thermal tuning parameter, such as pressure, chemical substitution, or magnetic field. Moreover, while there have been studies of the evolution of the critical fields $H_{c1}$[18] and $H_{c2}$[19,20] across putative QCPs, the role of associated fluctuations in destabilizing the vortex lattice at low $T$ has not been explored, despite the fact that they can have a profound influence on both the normal and SC state properties. In order to explore these questions, it is necessary to identify a material class that harbors both a QVL and a QCP somewhere within its tunable range of superconductivity. In this report, we consider one such candidate material and study the evolution of the low-$T$ vortex dynamics in the iron chalcogenide family FeSe$_{1-x}$X$_x$ where X = S, Te.

FeSe is unique among iron-based superconductors in that it develops nematic order, characterized by a spontaneous rotational symmetry breaking of the electronic state, without accompanying magnetic order[21–24]. With increasing $x$, the nematic phase transition is suppressed, terminating at a critical S (Te) concentration $x_c \sim 0.17$ (0.50), respectively[25,26]. Superconductivity persists across the entire series with $T_c$ peaking in the S-substituted family around 10 K for $x \approx 0.10$, where the spin fluctuations are also enhanced[27], and around 14 K in the Te-substituted family near the nematic end point[20,28]. In FeSe$_{1-x}$S$_x$, the magnitude of the SC gap $\Delta$ roughly halves outside of the nematic phase[29,30], suggesting that nematicity has a profoundly distinct influence on the SC properties of the two systems. Finally, the low

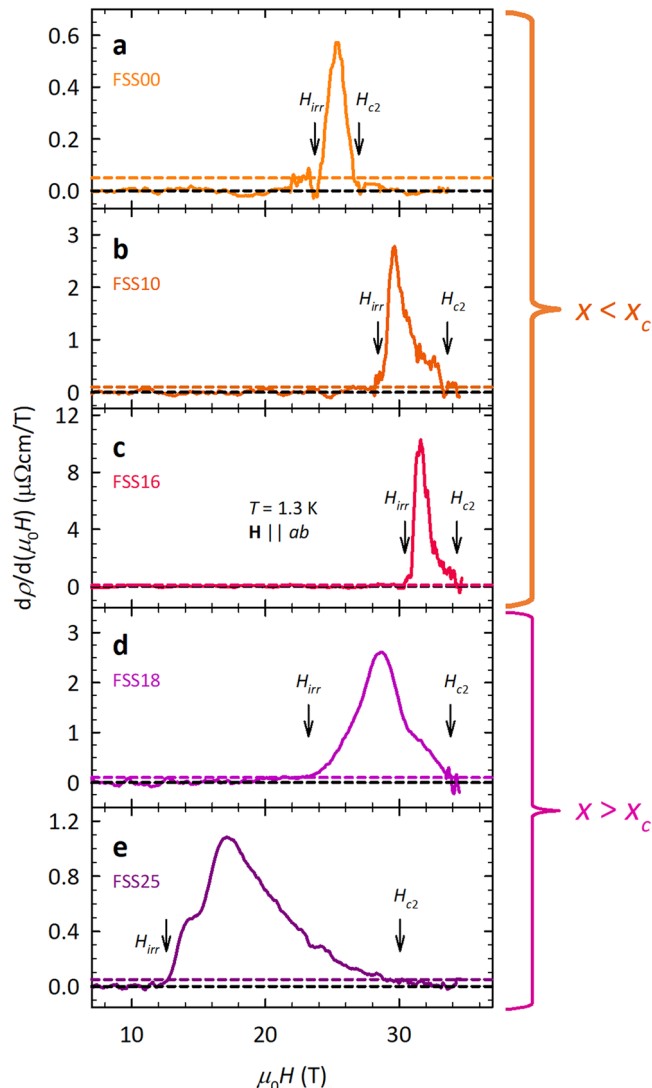

**Fig. 2 | Determination of $H_{irr}$ and $H_{c2}$ in FeSe$_{1-x}$S$_x$ with H∥$ab$ at $T$ = 1.3 K.** Field derivatives of the MR curves for **a** FSS00, **b** FSS10, **c** FSS16, **d** FSS18 and **e** FSS25. For all samples, $H_{irr}$ is determined from the field value above which d$\rho$/d($\mu_0H$) rises above a threshold−indicated by the colored horizontal dashed lines−set by the noise level, while $H_{c2}$ is the field value beyond which d$\rho$/d($\mu_0H$) returns to a constant value within the noise floor. The noise threshold is 0.05 $\mu\Omega$cm/T for FSS00 and FSS25 and 0.1 $\mu\Omega$cm/T for FSS10, FSS16, and FSS18, respectively. The marked expansion of the low-$T$ vortex liquid regime for $x > x_c$ is clear.

magnetotransport measurements down to $T/T_c \approx 0.03$ and in fields up to 60 T. These measurements cover a wide range of concentrations $x$ spanning the nematic QCPs in both systems and for FeSe$_{1-x}$S$_x$ in both longitudinal and transverse field orientations. Our study reveals an expanded QVL regime over a broad region of the cojoined FeSe$_{1-x}$X$_x$ phase diagram. Subsequent analysis indicates that the extent of the QVL regime is directly correlated with the strength of the SC order parameter, being anomalously broadened in the concentration ranges where $T_c$, $H_{c2}$, and/or the SC energy gap $\Delta$ are suppressed. By extending the study across two families, we can rule out a unique or dominant role for nematic or spin fluctuations, critical or otherwise, as well as for the BCS-BEC crossover in creating the QVL. Nevertheless, it is likely that their combined presence, coupled with the low dimensionality, creates the necessary conditions for quantum SC fluctuations to have a profound destabilizing effect on the VS, one that is amplified wherever the intrinsic superconductivity is weakened.

## Results

In this section, we will focus predominantly on the FeSe$_{1-x}$S$_x$ study, for which the bulk of the measurements were performed, and will summarize the corresponding findings for FeSe$_{1-x}$Te$_x$ in the "Discussion" section that follows. The key parameters for quantifying the extent of the QVL regime are $H_{irr}(0)$ and $H_{c2}(0)$, which are determined from magnetoresistance (MR) measurements using a low current excitation in order to stay within the $j \to 0$ limit where $H_{irr}(T) \approx H_m(T)$ (see Supplementary Note 1 where we also discuss the current dependence of $H_{irr}(0)$). Note that we use the same criteria for determining $H_{irr}(T)$ and $H_{c2}(T)$ for both families.

For the longitudinal field orientation (H∥$ab$), the MR in FeSe$_{1-x}$S$_x$ is negligible for $x \geq 0.16$ and small and linear for $x < 0.16$[32], making the identification of $H_{c2}(T)$ relatively straightforward. Here, we define $H_{c2}(T)$ as the field at which d$\rho$/d($\mu_0H$) returns to a constant value, i.e., when SC fluctuations are fully suppressed, and the MR assumes its normal state form, while $H_{irr}(T)$ is defined as the field above which the derivative d$\rho$/d($\mu_0H$) rises above a threshold set by the noise level.

Panels a–e of Fig. 2 show d$\rho$/d($\mu_0H$) for H∥$ab$ at a fixed temperature $T$ = 1.3 K for FeSe$_{1-x}$S$_x$ crystals with nominal $x$ values of 0, 0.1, 0.16, 0.18, and 0.25 (labeled hereafter FSS00, FSS10, FSS16, FSS18, and FSS25), respectively. The noise floor for each panel is given in the Figure caption and indicated by the colored horizontal dashed lines. The vertical black arrows in panels a-e indicate the corresponding values for $H_{irr}$ and $H_{c2}$. A marked expansion of the QVL regime is clearly evident beyond $x_c \sim 0.17$ that is due predominantly to a marked reduction in $H_{irr}$ rather than an enhancement in $H_{c2}$. This observation contrasts with a recent pressure study on FeSe$_{1-x}$S$_x$ ($x$ = 0.11)[39], where the expansion of the VL regime across the nematic QCP occurs due to an enhancement in $H_{c2}$.

In the transverse field orientation, the MR curves exhibit a complex $H$, $T$ dependence[33] that makes a determination of $H_{c2}(T)$ more challenging. Nevertheless, as shown in Supplementary Notes 2 and 3, by normalizing the field derivatives at each temperature and comparing them with the corresponding Hall response, a robust estimate of $H_{c2}(T)$ could be obtained for each MR curve measured (with the exception of FSS00−see Supplementary Note 2 for details).

Resultant $H_{irr}(T)$ and $H_{c2}(T)$ curves for H∥$ab$ are plotted in panels a–c of Fig. 3 for FSS00, FSS16, and FSS25, respectively. In all cases, $H_{irr}(T)$ lies well below $H_{c2}(T)$ at intermediate $T$, indicating the presence of a broad VL regime that narrows with decreasing temperature. In FSS00, there is a small but sharp change of slope in $H_{irr}(T)$ below $T \sim 1$ K that has been attributed previously to the emergence of a Fulde−Ferrell−Ovchinnikov−Larkin phase[40]. For FSS16, $H_{irr}(T) \approx 0.8$-$0.9$ $H_{c2}(T)$ as $T \to 0$, while for FSS25, $H_{irr}(0) < 0.5H_{c2}(0)$, as demonstrated in Fig. 2e.

Resultant $H_{irr}(T)$ and $H_{c2}(T)$ curves for H∥$c$ are plotted in panels e-g of Fig. 3 for FSS13, FSS16, and FSS25, respectively. In contrast to

carrier density coupled with its relatively high $T_c$ has led to speculation that superconductivity in FeSe and its cousins sits proximate to a Bardeen−Cooper−Schrieffer Bose-Einstein condensate (BCS−BEC) crossover[24].

The normal state transport properties of FeSe$_{1-x}$S$_x$ exhibit many features synonymous with quantum criticality and non-Fermi-liquid behavior[31–37]. In particular, the limiting low-$T$ resistivity $\rho(T)$, exposed by the application of a large magnetic field, traces out a fan-like region of $T$-linearity centered on the nematic QCP[31,32,36] below which a $T^2$ resistivity is recovered, with a coefficient that is strongly enhanced upon approaching the QCP (from the high-$x$ side)[38]. In FeSe$_{1-x}$Te$_x$ near the nematic QCP, there is evidence for a poorly metallic or incoherent normal state that may also indicate a non-Fermi-liquid ground state though high-field, low-$T$ transport measurements have not yet been reported.

In this report, we investigate the evolution of the vortex state in a series of FeSe$_{1-x}$S$_x$ and FeSe$_{1-x}$Te$_x$ single crystals via detailed

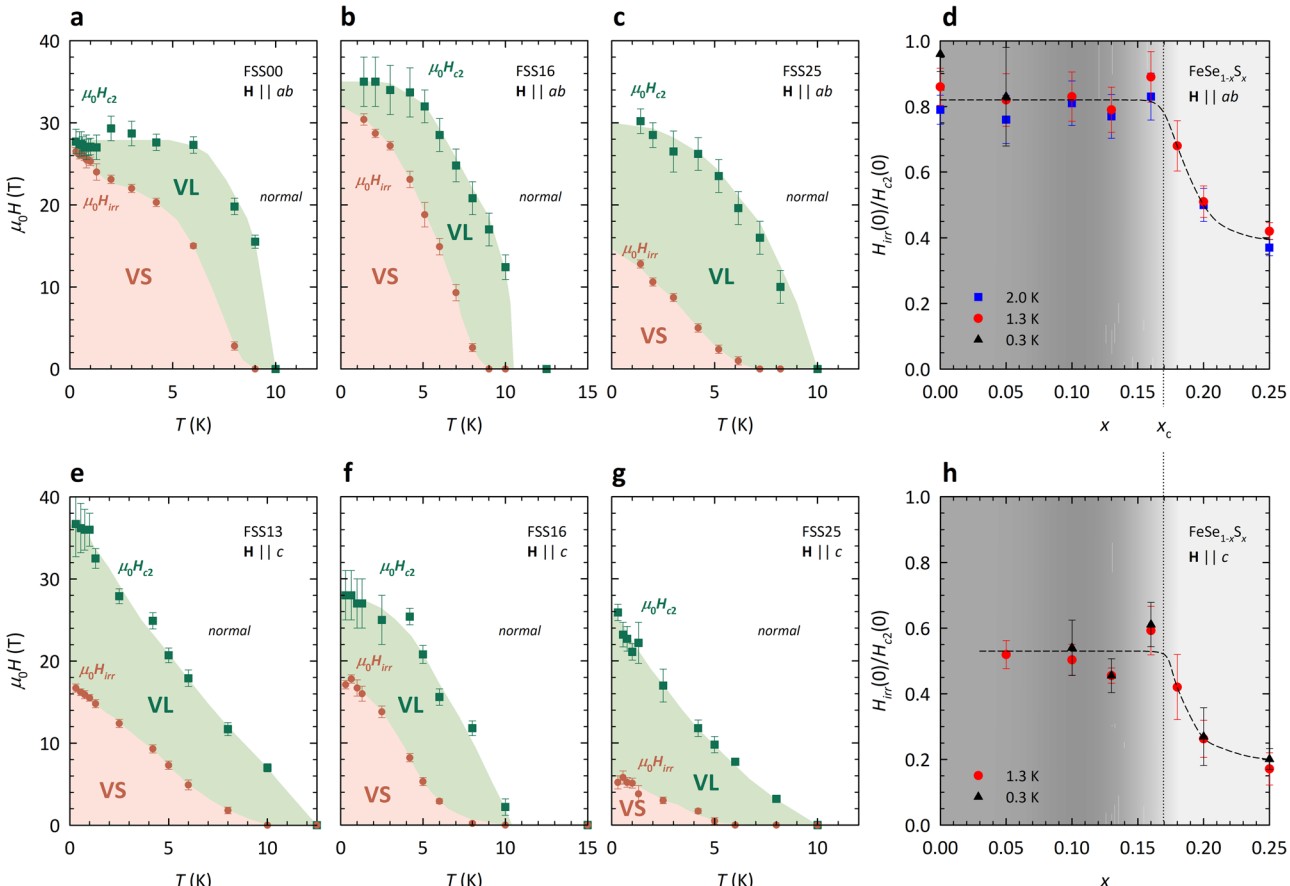

**Fig. 3 | Paradigmatic $H$–$T$ phase diagrams of FeSe$_{1-x}$S$_x$ in longitudinal and transverse magnetic fields.** $H_{irr}(T)$ (orange circles) and $H_{c2}(T)$ (green squares) curves extracted from the MR derivatives for **a** FSS00, **b** FSS16, and **c** FSS25 with **H**∥$ab$ and for **e** FSS13, **f** FSS16, and **g** FSS25 with **H**∥$c$. The VL state is shaded green. The Meissner phase at low fields has been omitted for clarity. Error bars for $H_{irr}$ and $H_{c2}$ are estimated from the threshold noise level shown, for example, in Fig. 2. The larger error bars for $H_{c2}$ at low $T$ in **b** are due to the limited field range of our experiment. (For this reason, $H_{c2}$ could not be determined below 1.3 K in all

samples). Larger error bars for $H_{c2}$ at low $T$ in panels **e** and **f** are due to the emergence of quantum oscillations as indicated in Supplementary Fig. 2. **d**, **h** The ratio $H_{irr}(0)/H_{c2}(0)$ deduced at the temperatures indicated for **H**∥$ab$ and **H**∥$c$, respectively. In both panels, the black dashed line is a guide to the eye, while the dark (light) gray backgrounds refer to the nematic (non-nematic) phases. The vertical dotted line spanning panels **d** and **h** locates the position of the nematic QCP at $x = x_c \approx 0.17$. Finally, the error bars in panels **d** and **h** are composite errors obtained from the errors in $H_{irr}$ and $H_{c2}$.

what is found in parallel fields, $H_{irr}(T)$ is suppressed relative to $H_{c2}(T)$ down to the lowest temperatures at all concentrations. It is also evident that $H_{c2}(T)$ follows a distinct $T$ dependence in the two field orientations, which for **H**∥$c$ does not follow the standard Ginzburg–Landau expression $H_{c2}(T) = H_{c2}(0)(1 - (T/T_c)^2)$. This non-standard $T$-dependence is likely to reflect the multi-band character of FeSe$_{1-x}$S$_x$, though it may also result from proximity to the BCS–BEC crossover regime[41,42] (to be discussed in more detail below).

In order to quantify the extent of the VL regime, we define $H_{irr}(0)/H_{c2}(0)$ as the ratio of the two field scales as $T \to 0$ (determined at the lowest measured temperatures–typically 0.3 K or 1.3 K). The evolution of $H_{irr}(0)/H_{c2}(0)$ with $x$ is shown in Fig. 3d for **H**∥$ab$ at three representative temperatures (0.3, 1.3, and 2.0 K) and in Fig. 3h for **H**∥$c$ at $T = 0.3$ K and 1.3 K. The overall trend is clear and appears to be independent of temperature. (Two other procedures for the determination of $H_{c2}(T)$, presented in Supplementary Fig. 5, demonstrate a qualitatively similar evolution of the ratio with $x$.) For **H**∥$c$, $H_{irr}(0)/H_{c2}(0) \leq 0.6$ for all finite $x$. The most striking feature, however, is the marked, twofold drop in $H_{irr}(0)/H_{c2}(0)$ at high $x$, revealing an expansion of the QVL phase beyond $x_c$ in both field orientations. This is the main finding of the FeSe$_{1-x}$S$_x$ study. The corresponding result for FeSe$_{1-x}$Te$_x$ is shown together with FeSe$_{1-x}$S$_x$ in Fig. 4 and will be discussed in the following section.

## Discussion

According to Ginzburg–Landau theory, the $H$–$T$ phase diagram of a type-II superconductor depends on three fundamental parameters: (i) pinning strength, (ii) thermal fluctuations, and (iii) quantum fluctuations (see Fig. 1). The pinning strength represents the effective energy barrier a vortex has to overcome in order to become mobile. It can be evaluated from the ratio $j_c/j_0$ of the depinning current density (above which the vortices start to move) and depairing current density (above which Cooper pairs are destroyed)[1]. Magnetization measurements in pure FeSe give $j_c \approx 4 \times 10^4$ A/cm$^2$ [43], while $j_0 = 4\bar{H}_c/(3\sqrt{6}\lambda)$, where $\bar{H}_c$ is the thermodynamic critical field and $\lambda$ is the London penetration depth. Taking $\mu_0\bar{H}_c(0) = 0.21$ T[44] and $\lambda(0) \approx 400$ nm[45], we obtain $j_c \approx 2 \times 10^7$ A/cm$^2$ and $j_c/j_0 \approx 2 \times 10^{-3}$, a value similar to that found in cuprates and ~10–100 times smaller than in conventional superconductors[1]. Such a value suggests that pinning in FeSe is weak so that the melting transition between the VS and the VL phase will be only weakly perturbed by the presence of disorder[1]. According to ref. 43, however, $j_c$ in FeSe$_{1-x}$S$_x$ follows a similar power-law decay $H^{-0.5}$ to that observed in iron-pnictides and attributed to strong pinning by sparse nm-sized defects[46]. Nevertheless, it seems reasonable to assume that at the current densities applied in this work $F_L < F_p$ and that $H_{irr}(T) \approx H_m(T)$, i.e., a finite resistivity is unambiguously related to the presence of a VL phase (see Supplementary Note 1). This is especially

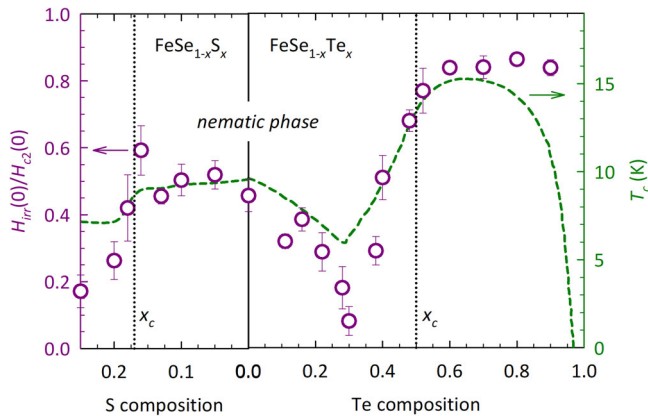

**Fig. 4 | Comparison of the QVL regimes in FeSe$_{1-x}$S$_x$ and FeSe$_{1-x}$Te$_x$ for H∥c.** Symbols represent the ratio $H_{irr}(0)/H_{c2}(0)$ deduced at the lowest temperatures $T = 0.3$ K for FeSe$_{1-x}$S$_x$ and $T = 0.6 - 1.5$ K for FeSe$_{1-x}$Te$_x$. The data for FeSe$_{1-x}$S$_x$ are taken from Fig. 3h and for FeSe$_{1-x}$Te$_x$ from a similar analysis of data from our previous study[20] (see Supplementary Note 5). The green dashed line is the $x$-evolution of the corresponding SC transition temperature $T_c$ which for FeSe$_{1-x}$Te$_x$ is reproduced from Fig. 3 in ref. [20]. Vertical dotted lines indicate the positions of the nematic QCPs. The error bars are composite errors obtained from the errors in $H_{irr}$ and $H_{c2}$.

true for the longitudinal orientation **H**∥*ab* where **j**∥**H** and, therefore $F_L = 0$.

The strength of thermal fluctuations is usually quantified by the Ginzburg number $G_i = (1/8\pi\mu_0) \times ((k_B T_c \Gamma)/(H_{c2}^2(0)\xi_\parallel^3(0)))^2$ —itself, a measure of the relative size of the thermal and condensation energies within a coherence volume[1,47]. Here, $k_B$ is the Boltzmann constant, $\Gamma = \xi_\parallel(0)/\xi_\perp(0)$ is the anisotropy of $H_{c2}$, and $\xi_\parallel(0)$ and $\xi_\perp(0)$ are the in- and out-of-plane coherence lengths at 0 K, respectively. From existing thermodynamic data, $G_i$ in FeSe is estimated to be $5 \times 10^{-4}$ [44], which is lower than in the cuprates where thermal fluctuations are very strong ($10^{-3} < G_i < 10^{-1}$)[48], but around four orders of magnitude larger than in classical superconductors[1]. Indeed, evidence of strong thermal SC fluctuations has been found in several normal state properties for pure FeSe[41], as well as for FeSe$_{1-x}$S$_x$ beyond $x_c$[49], in line with the broad VL regime found here (see Fig. 3).

As $T \to 0$, the effect of thermal fluctuations inevitably diminishes, leaving only quantum fluctuations of the SC order parameter to influence the vortex dynamics. The persistence of a broad QVL regime in Fig. 3 thus supports the notion that the vortex lattice in FeSe$_{1-x}$S$_x$ is destabilized at very low temperatures by strong quantum fluctuations inherent in the system. According to the Lindemann criterion, quantum melting occurs when the zero-point amplitude of the vortices becomes an appreciable fraction (typically 0.1–0.2) of the vortex spacing. At that point, there are strong fluctuations in the positions of the vortex lines, i.e., fluctuations in the phase field of the SC order parameter, which leads to the formation of a QVL. (The corresponding amplitude fluctuations can be ignored since they are usually confined to a narrow region around $T_c$ and $H_{c2}$.)

The strength of such quantum fluctuations can be estimated by the dimensionless quantum resistance $Q_u = e^2/(\hbar\eta s)$, where $e$ is the elementary charge, $\hbar$ the reduced Planck constant, $\eta$ the friction coefficient, and $s = 5.5$ Å the interlayer spacing[5]. Using the conventional BCS approach in which the friction coefficient $\eta$ is proportional to the normal state conductivity $1/\rho_n$[1,5,6], we obtain a small value of $Q_u$ for pure FeSe. Theories that go beyond the mean-field approach[50,51], however, suggest that the friction coefficient can be drastically reduced due to details of the electronic structure and/or proximity to other non-SC orders, leading to a significant enhancement of $Q_u$ in line with the broad QVL regime observed here.

With this in mind, let us now turn to consider the QVL regime in more detail. The two-fold drop in $H_{irr}(0)/H_{c2}(0)$ beyond $x_c$ indicates a marked broadening of the QVL beyond the nematic QCP. Thermal conductivity and specific heat measurements in FeSe$_{1-x}$S$_x$[30] indicate a similarly abrupt change in the SC gap structure at $x_c$ while scanning tunneling spectroscopy[29] detected a twofold drop in $\Delta$ beyond $x_c$[29], in remarkable agreement with the $x$-evolution of $H_{irr}(0)/H_{c2}(0)$. Such a correspondence between $H_{irr}(0)/H_{c2}(0)$ and $\Delta$ clearly demonstrates an intimate relationship between the extent of the QVL regime and the strength of superconductivity in FeSe$_{1-x}$S$_x$.

This correspondence becomes even more evident when we compare these ratios for the two chalcogenide families, as is done in Fig. 4. The most striking feature of this plot is that the marked downturns in $H_{irr}(0)/H_{c2}(0)$ (purple circles)−corresponding to an expansion of the QVL regime−coincide with a dip in their respective $T_c(x)$ curves (green dashed line). This correlation provides strong evidence that the QVL regime is the most extended wherever the intrinsic superconductivity is the weakest. It should be stressed, however, that this is not a trivial finding since such an expanded QVL is never observed in conventional but weakly coupled superconductors. This then begs the question: what is the cause of strong SC quantum fluctuations and associated QVL phase that permeate the chalcogenide phase diagram?

As already mentioned, SC quantum fluctuations can be significantly enhanced due to proximity to a non-SC order[50,51], fluctuations of which can cause an additional scattering of quasiparticles within the vortex cores[52]. In underdoped La$_{2-x}$Sr$_x$CuO$_4$, for example, a QVL phase is formed close to the antiferromagnetic (AFM) endpoint, where spin fluctuations are known to be prevalent[16]. In FeSe$_{1-x}$S$_x$, although no magnetic order exists at ambient pressure, strong AFM fluctuations are present[53–55], associated with the static order that develops under applied pressures[56]. Such fluctuations could affect the vortex dynamics and possibly establish the QVL state. With increasing $x$, however, AFM fluctuations in FeSe$_{1-x}$S$_x$ are progressively suppressed and become negligible beyond the nematic QCP[27,57,58], exactly where the QVL state seems to be the most pronounced. Although a detailed Te composition dependence of spin fluctuations for $0 \le x \le 0.50$ is currently lacking, a related pressure study showed that magnetic interactions do become weaker in the higher-$T_c$ dome region[59]. Thus it seems unlikely that spin fluctuations can account for the development of the QVL phase in either system.

Strong signatures of nematic quantum criticality have been seen in the elasto-resistivity, Hall and MR responses in FeSe$_{1-x}$S$_x$[25,32,33,35,37]. Nematic fluctuations diverge as $x \to x_c$[25], while the fan of $T$-linear resistivity and enhanced $T^2$ coefficient (on approach to $x_c$ from the high $x$ side[38]) have both been attributed to enhanced quasiparticle scattering off critical nematic fluctuations that may also enhance vortex dissipation at low-$T$ beyond $x_c$. The absence of any sharp dip in $H_{irr}(0)/H_{c2}(0)$ near the QCP in FeSe$_{1-x}$Te$_x$, however, coupled with the fact that the QVL regime is most pronounced inside the nematic phase (Fig. 4), appears to rule that possibility out. This latter point also rules out the loss of orthorhombicity and the associated disappearance of twin boundaries as pinning centers as the origin for the expanded QVL regime in FeSe$_{1-x}$S$_x$[60,61].

In pure FeSe, the ratio $\Delta/\epsilon_F \approx 0.3 - 1.0$, where $\epsilon_F$ is the Fermi energy[24,42,45], placing FeSe close to the BCS-BEC crossover (see Supplementary note 6). Proximity to this crossover will inevitably lead to strong SC fluctuations and potentially to an enhancement of both the VL and QVL regimes. ARPES measurements on FeSe$_{1-x}$S$_x$[62] claim to show strong evidence for BEC-like superconductivity beyond $x_c$, while recent thermodynamic studies have revealed that the shape of the heat capacity jump at $T_c$ changes across $x_c$, exhibiting non-mean-field behavior that is reminiscent of a BEC transition[49]. Certainly, the marked drop in $H_{irr}(0)/H_{c2}(0)$ across $x_c$ shown in Fig. 3d, h seems to correlate with enhanced SC fluctuations associated with BCS−BEC crossover[49,62]. Again, the comparison with FeSe$_{1-x}$Te$_x$ turns out to be

highly instructive. In the latter, the BCS−BEC crossover is most pronounced at $x \approx 0.52$, i.e., where $H_{irr}(0)/H_{c2}(0)$ reaches almost the highest value, and the QVL regime is least pronounced. This suggests that proximity to the BCS−BEC crossover while creating an ideal environment for strong SC fluctuations to proliferate, cannot be the dominant reason for an expanded QVL regime in the chalcogenides. Nevertheless, given the profound influence that spin fluctuations, nematic fluctuations, and the putative BCS−BEC crossover appear to have on the normal state, it is highly likely that they play some role in the mixed state (vortex) dynamics of FeSe$_{1−x}$X$_x$.

The finding that the drop in $H_{irr}(0)/H_{c2}(0)$ in FeSe$_{1−x}$S$_x$ coincides with a similar reduction in $\Delta$ arguably provides the key to unraveling the origin of the expanded QVL regime beyond $x_c$. Generally[63], pinning strength is determined by the product of the condensation energy $U_c$ ($\propto \Delta^2$) and the minimum volume change associated with vortex motion ($\propto \xi^3$). Hence, the twofold reduction in $\Delta$ beyond $x_c$[29] must have a sizeable effect on the overall pinning strength and, thus, on the stability of the vortex liquid, i.e., the melting line. While a comparable study of $\Delta(x)$ in the Te-substituted family has not yet been reported, the correlation between the expanded QVL regime and the reduced $T_c$ and $H_{c2}(0)$ suggests that a similar relation between $H_{irr}(0)/H_{c2}(0)$ and $\Delta$ may exist in FeSe$_{1−x}$Te$_x$ too. Finally, we remark that the possible influence of the so-called Bogoliubov Fermi surface[64], invoked to account for the high residual density of states found beyond $x_c$ in FeSe$_{1−x}$S$_x$[30], should also be considered.

Before closing, we turn to consider why the QVL phase is much reduced, if present at all, for $x < x_c$ in parallel fields. In this field orientation, vortices will be configured to lie between the SC planes, and while the SC state in FeSe$_{1−x}$S$_x$ is not strongly anisotropic, they may still become susceptible to an additional pinning potential created by the lamellar nature of the crystal. This in turn, will raise the overall barrier height, thereby reducing the propensity for quantum fluctuations to destabilize the lattice. Beyond $x_c$, however, even this additional barrier becomes ineffective.

In summary, by carrying out a comparative transport study of the mixed state in both FeSe$_{1−x}$S$_x$ and FeSe$_{1−x}$Te$_x$, we have uncovered two markedly enhanced QVL regimes that are concomitant with a reduced SC pairing strength. The persistence of this broad QVL regime over such an extended region of the phase diagram, however, suggests that key elements of the electronic state of iron chalcogenides, such as the presence of strong nematic and magnetic fluctuations and proximity to the BCS−BEC crossover, conspire to destabilize the VS to such an extent that any further weakening of the SC order leads to the QVL regime being amplified. Whatever the origin of this reduced SC strength beyond $x_c = 0.17$ in FeSe$_{1−x}$S$_x$ and at $x \approx 0.30$ in FeSe$_{1−x}$Te$_x$, the existence of such an extended QVL regime provides an unprecedented opportunity to study the dynamics of this elusive state in great detail. At the same time, it enables us to identify new guiding principles for its emergence that go beyond the simple requirement of an SC state of reduced dimensionality, specifically the propensity to fluctuating order and proximity to a BCS-BEC crossover. While in amorphous films, the disorder is likely to be playing a key role, in the underdoped cuprates and organic superconductors, fluctuating order is also present. Thus, from a theoretical perspective, the results reported here should help to motivate future investigations into how strong order parameter fluctuations act to destabilize the vortex lattice in unconventional superconductors.

## Methods

FeSe$_{1−x}$S$_x$ ($0 \leq x \leq 0.25$) and FeSe$_{1−x}$Te$_x$ ($0 \leq x \leq 0.48$) crystals were grown by the chemical-vapor-transport (CVT) technique, while FeSe$_{1−x}$Te$_x$ crystals with $0.52 \leq x \leq 0.90$ were obtained by the Bridgman method. For the latter, the Te annealing procedure was applied to minimize excess Fe[65]. The actual Te composition $x$ of crystals synthesized by the CVT method is determined for each sample from the $c$-axis length measured by X-ray diffraction. The $x$ values of the FeSe$_{1−x}$S$_x$ crystals, as well as the FeSe$_{1−x}$Te$_x$ crystals grown by the Bridgman method, are taken from the nominal values. The MR measurements on FeSe$_{1−x}$S$_x$ and FeSe$_{1−x}$Te$_x$ were performed at the High Field Magnet Laboratory (HFML) and the International MegaGauss Science Laboratory at the University of Tokyo, respectively. The orientation of the samples was determined by using a Hall probe mounted on the rotating sample platform. All measurements were performed at fixed temperatures while sweeping the field from $-H_{max}$ to $+H_{max}$. Complementary Hall effect measurements were also performed at HFML in magnetic fields up to 33 T and at fixed temperatures down to 0.3 K. At each temperature, the transverse MR signal $V_x$ (Hall signal $V_y$) was symmetrized (antisymmetrized) in order to eliminate any finite Hall (MR) component. Analysis of the Hall data is shown in Supplementary Note 3, while additional sample details are given in Supplementary Note 7.

## Data availability

The data that support the plots within this paper and other findings of this study are available from the University of Bristol data repository, data.bris, at https://doi.org/10.5523/bris.1c3xho5w6nfbf2m7kjwyh50x99.

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

## Acknowledgements
The authors acknowledge enlightening discussions with M. Berben, P. Chudzinski, C. Duffy, and R. D. H. Hinlopen.

## Author contributions
N.E.H., T.S., Y.M., and S.L. conceived the project while N.E.H., M.Č., and K.I. supervised it. S.K. synthesized and characterized the single crystals of $FeSe_{1-x}S_x$. M.W.Q. and M.S. synthesized the single crystals of $FeSe_{1-x}Te_x$ for $x < 0.48$ and K.I., M.W.Q., Y.U., T.O., and T.W. for $x > 0.52$. Y.U., T.O., and T.W. performed Te-annealing of the as-grown $FeSe_{1-x}Te_x$ single crystals. M.Č., S.L., J.A., and J.B. carried out the high-field experiments on $FeSe_{1-x}S_x$, and K.M., K.I., S.I., and K.K. performed the high-field experiments on $FeSe_{1-x}Te_x$. Y.T.H. checked the dependence of the irreversibility field on applied current in $FeSe_{1-x}S_x$. M.Č. and S.L. analyzed the data. M.Č. and N.E.H. wrote the paper with input from all the other co-authors.

## Funding
The authors acknowledge the support of HFML at Radboud University, a member of the European Magnet Field Laboratory, the Netherlands Organization for Scientific Research (NWO) (Grant No. 16METL01) 'Strange Metals', the ERC under the European Union's Horizon 2020 research and innovation program (Grant Agreement no. 835279-Catch-22), EPSRC (Grant ref. EP/V02986X/1), Grants-in-Aid for Scientific Research (KAKENHI) Innovative Area "Quantum Liquid Crystals" (No. JP19H05824) from Japan Society for the Promotion of Science (JSPS) and by the Japan Science and Technology Agency (JST) CREST program (Grant No. JPMJCR19T5) and JSPS KAKENHI Grant Nos. JP19H00649, JP20K03849, JP21H04443, JP22H00105, JP22KK0036. The work at Hirosaki was supported by Hirosaki University Grant for Distinguished Researchers from fiscal years 2017–2018. JA acknowledges an EPSRC Doctoral Prize Fellowship (Ref. EP/T517872/1) and a Leverhulme Early Career Fellowship.

## Competing interests
The authors declare no competing interests.
