## [Peer Review File · Nature Communications]

REVIEWER COMMENTS

Reviewer #1 (Remarks to the Author):

In their present manuscript, Culo, Licciardello et al. discuss the evolution of the two critical magnetic fields H_{irr} and H_{c2} of the normal-to-superconducting transitions in the iron-based superconductor Fe(Se,S). They report a significant separation of H_{irr} and H_{c2} towards $T \rightarrow 0K$ across the nematic quantum phase transition which they interpret as a signature of an enhanced Quantum Vortex Liquid regime. They discuss possible reasons and implications for its formation / stabilization, such as the role of AFM and nematic quantum fluctuations and the proximity of FeSe to a BEC-BCS crossover. However, from their data alone they cannot identify the dominant driving force.

As such, I feel that the present results are certainly rather interesting, but I do not see how they will advance the wider field of research. Instead, they quite strongly relate to Fe(Se,S) only. As such, I cannot recommend publication in Nature Communications but would suggest a more specialized journal.

Additionally, I have the following remarks to the authors:

Relation to previous work:

A broadening of the superconducting transition width in the resistivity for $H \parallel c$ was previously reported across the nematic QCP under pressure (<https://journals.aps.org/prl/abstract/10.1103/PhysRevLett.127.246402>). In fact, the derivatives in Fig.2 (present manuscript) and Fig.4 (reference above) appear very similar, suggesting they describe the same effect. In contrast to the present paper, the authors above related the broadening directly to the nematic QCP since the transitions become sharp again towards higher pressures. Moreover, the authors above interpreted the temperature dependence of the resistivity between H_{irr} and H_{c2} as evidence for a vortex liquid which would freeze to a vortex glass at zero temperature. Since both reports cover the same temperature and field range, this suggests to rule out a QVL in this regime, plus the pressure vs chemical composition dependence seem to differ outside the nematic phase. Could the present authors comment how their results compare to the reference above?

Current dependence:

In the Introduction, the present authors argue that they work in the zero-current limit $j \rightarrow 0$. However, at the lowest temperatures, a previous study e.g. on YBCO demonstrated that H_{irr} may strongly depend on the applied current (<https://www.pnas.org/doi/pdf/10.1073/pnas.2021216118>) even though for this system, a very large critical current could be assumed. Did the present authors check for a current dependence of their results to support their claim of working in the low-current limit?

Relation to nematicity / interpretation of the data:

The present manuscript is part of an ongoing debate about the role of nematic quantum critical fluctuations on superconductivity and the normal state behavior of Fe(Se,S), as set out in the manuscript at several locations. However, for a precise localization of the nematic QCP, the exact sulfur content of Fe(Se,S) samples was pointed out to be crucial in Ref. [39]. Similar to their previous works (Refs. [35,36,40,41]), the present authors use the nominal sulfur concentrations instead of the EDX values to locate their samples in the phase diagram. Yet, I recall that Ref. [39] claimed that sample FSS20 still showed a nematic transition temperature. In this case, the drop in H_{irr}/H_{c2} would occur well inside the nematic phase, and not across its boundary, and hence it could be related to strong AFM fluctuations. Note for example that Xiang et al. reported the emergence of a magnetic phase inside(!) the nematic phase (<http://link.aps.org/doi/10.1103/PhysRevB.96.024511>). Could the present authors clarify the exact location of the nematic phase boundary based on the presently used samples and/or the values of T_s and/or the measured x_{EDX} ? The chemical vapor transport-grown samples are known to show a large scattering of real sulfur contents due to the thermal gradients and such, a per-sample assessment is required.

Reviewer #2 (Remarks to the Author):

The manuscript reports the magnetic field v.s. temperature phase diagrams determined based on the resistivity data of the iron-based weakly-anisotropic three-dimensional (3D) superconductors FeSe_{1-x}S_x. The authors find that, as x is increased so that the nematic order is lost, an unusually broad quantum vortex liquid (QVL) regime is present at least in the temperature range higher than $0.03 T_c$. Based on the fact that a measurable width of the QVL regime appears even in the nematic ordered samples with $x < x_c$, the authors seem to have concluded that the origin of the expansion of the QVL regime is not simply due to the highly enhanced superconducting (SC) fluctuation in the materials with $x > x_c$ but rather due to a combined effect of the SC fluctuation, spin fluctuation in $x < x_c$, and the loss of the nematic order in $x > x_c$.

The experimental fact showing the unusually broad QVL is intriguing and is worth publishing in this journal as one of strange phenomena in the novel superconductors rarely seen in conventional superconductors. However, the authors' interpretation on the origin of the broad QVL regime is inconsistent with other published works on the same family of materials, e.g., Ref.43. For instance, the phrases "enhanced SC fluctuation which are not related to vortex motion" on page 3 (line 87) and page 8 (line 223) and "the superconducting quantum fluctuations are too weak" in Abstract are incomprehensible, because strong thermal fluctuations usually enhance the corresponding quantum fluctuations at finite temperatures. The present version of this manuscript cannot be accepted for publication, and a drastic revision of the manuscript is necessary to make the content of the research work readable.

(a) Obviously, the crucial error in their interpretation on the resistivity data consists in their estimation of the dimensionless quantum resistance Q_u in sec.3. There, they assume that the friction (damping) coefficient η (in the notation by Refs.1, 5, and 6) takes the conventional BCS form proportional to the normal conductivity $1/\rho_n$. The coefficient η is found as the

coefficient of the dissipative time-derivative term of the time-dependent Ginzburg-Landau equation following from the corresponding electronic hamiltonian describing the superconductor of one's interest. For unconventional superconductors, such theoretical studies are available;

PRB 66, 094513 ('02)

for high T_c cuprates and

PRB 105, 174510 ('22)

for FeSe ($x=0$) near $H_{c2}(0)$. In the latter, the vortex friction coefficient η is proportional to η_n there (eqs.19 and 20 there), while eqs.(26) -- (32) are related to this issue in the former. As seen in these works, the friction coefficient is drastically reduced, i.e., the quantum SC fluctuation is enhanced irrespective of the thermal fluctuation, due to electronic details and/or an effect of other (if any) nonSC orders. Namely, the friction coefficient may be independent of the normal resistivity ρ_n .

In relation to this, the author should understand the following things on how to theoretically describe possible symmetry breakings and orders: The QVL is the state created by strongly interacting SC fluctuations, and, as far as other possible orderings do not occur within the QVL regime, effects of such other orders or fluctuations are incorporated in the corresponding Ginzburg-Landau free energy F_{GL} expressed by the SC order parameter in a form affecting the coefficient of each term of F_{GL} . Namely, in any case, the broader QVL is a consequence of thermal and/or quantum SC fluctuation enhanced by the decrease and disappearance of the nematic order. In fact, the heat capacity data of FeSe_{1-x}S_x in Ref.43 have clearly shown the SC fluctuation enhanced drastically with increasing x . It might be possible that the nematic or other quantum critical fluctuation near x_c reduces the SC fluctuation effect. In fact, the authors mention in relation to Refs.19 and 20 that the H_{c2} is enhanced, implying an enhanced SC order. The resulting reduction of the SC fluctuation competes with the effect of enhancing the SC fluctuation due to the disappearance of the nematic order. In relation to the sentence on the line 149 on page 7, it might be possible that H_{irr}/H_{c2} in $x < x_c$ is nearly constant and has not decreased with increasing x as a consequence of this competition.

Further, the authors mention that the vortex pinning is enhanced with increasing x . On the other hand, when an electron system becomes dirtier with increasing x , conventionally the strength of the SC fluctuation tends to be enhanced as a consequence of the reduction of the superfluid density.

To summarize, the authors need to revise their explanation on the expansion of the QVL regime.

(b) It seems that the references to be quoted should be reconsidered. Experiments which have led to Ref.10 have been performed in relation to the theoretical review paper treating the 2D case,

1) Int. J. Mod. Phys. 10, 601 ('96).

In relation to this, I notice that Ref.11 is absent (The title is incorrect).

Further, Ref.14 has been summarized based on a support via another theoretical paper treating both the 2D and 3D cases,

2) J. Phys. Soc. Jpn. 72, 2930 ('03) (arXiv:cond-mat/0210626)

(In particular, see Figs.3, 10, and 11 there).

In these two papers, the resistivity has been examined as a measure of the quantum fluctuation in QVL. In contrast, Refs.7 and 8 treat the QVL in a 2D model assuming nondissipative dynamics and thus, are not comparable with any resistivity data.

(c) When an expert on the vortex matter in superconductors starts reading this manuscript, he would stop reading the manuscript further by looking at Fig.1A and related sentences on pages 1 and 2, because Fig.1A is completely erroneous. The ordered phase is the unpinned vortex solid (VS). The unpinned VS known as the Abrikosov lattice is not a VL but an anisotropic SC phase. The VL is induced by the SC fluctuation and is one part of the normal phase in which the resistivity is nonvanishing for any direction of the current, while the resistivity in a current parallel to the magnetic field is zero in the unpinned VS. Further, the authors possibly assumed that the pinning-free limit of real VS is the unpinned VS. At least, in the thermodynamic limit, the H_{irr} or the vortex-glass transition field in the pinning-free limit of a real superconductor approaches the vortex lattice melting line $H_m(T)$ and does not decrease down to a value below H_m . The resulting VS is called as Bragg-glass.

See PRB 43, 130 (1991), J.Phys.Soc.Jpn.65, 3998 ('96), and PRB 52, 1242 ('95) for details.

At least, the figure A needs to be deleted from Fig.1 or to be replaced by a theoretically valid one, and the final sentence on page 1 must be deleted.

Strictly speaking, the figure D of Fig.1 is also erroneous: In 3D case at zero temperature, the QVL is absent, i.e., $H_{irr} = H_m$, although the field range of VL is broadened at very low but finite temperatures. This theoretical picture follows by directly applying the analysis in the seminal theoretical work, PRB 31, 7124 ('85), to the zero temperature case. Therefore, I respect that the phrase "even at temperatures as low as $0.03 T_c$ " had been written in sec.4 in the manuscript. If possible, it would be better that the authors also revise Fig.1D in this sense.

Through the paragraph below Fig.1, I feel that in VL $F_p < F_L$ is assumed. As mentioned above, however, the VL is one part of the normal phase, and thus, $F_p=0$ on average there. I feel that the authors consider here the pinning strength for a single vortex. However, the pinning strength for a single vortex is ill-defined in VL. In any case, the paragraph below Fig.1 has been written based on the old mean field picture that the H_{c2} line is the SC transition line. It is well known at present that this old picture is erroneous. In fact, based on this old picture, the nonvanishing resistivity in a current parallel to the field in the VL regime cannot be explained (see

J.Phys.Soc.Jpn.58, 1906 ('89)).

Finally, regarding the details of data, Fig.3A (i.e., $x=0$ and $H // ab$ case) seems to be different from that in Ref.44. Further, the authors have never discussed about an effect of the paramagnetic pair breaking which was one of the main issues in Refs.44 and 46. At least the H_{c2} curves in A,B, and C of Fig.3 become flat at lower temperatures possibly reflecting a strong paramagnetic pair-breaking. Discussing about a reason why the paramagnetic pair breaking becomes less effective with increasing x would be interesting. For instance, is it due to the fact that the system becomes dirtier with increasing x ?

Reviewer #3 (Remarks to the Author):

The paper by Culo et al. reports on the observation of an expanded quantum vortex liquid (QVL) regime in S-doped FeSe that is stabilized by quantum fluctuations close to the nematic quantum critical point. Their main experimental finding constitutes a drop of the ratio of irreversibility field to upper critical field at the nematic quantum critical point, based on an analysis of magnetoresistance measurements in high magnetic fields. If confirmed, this would certainly constitute an interesting result that would warrant publication in Nat. Comm. However, the present version of the paper indicates several issues that need to be addressed before further decisions can be made.

The main concern relates to the interpretation of the finding of the expanded QVL state in connection with the role of quantum critical degrees of freedom of magnetic/nematic origin. The authors themselves attest that the “absence of any sharp dip in H_{irr}/H_{c2p} near the QCP”, but do not take that into account when making connections to quantum criticality. Obviously, the step-like change that they observe does not meet the intuitive expectations of quantum critical fluctuations being important, but rather points to some discontinuous change of some of the superconducting parameters (e.g. like gap parameters). Therefore, all arguments regarding the importance of magnetic vs. nematic fluctuations to be important appear very speculative.

The second concern relates to the determination of H_{irr} and H_{c2} from the transport data. Even though the authors take a great effort in the main text and in the supplement to justify that their finding is independent of the choice of criterion, I still feel somewhat uncomfortable when seeing the absence of any feature at H_{c2_A} in Supplementary Figure 4 E and F. In the main text, the authors argue that the longitudinal magnetoresistance is negligible for high x , and therefore magnetoresistance is straightforward to analyse. If this is the case, then I would also expect that it should be straightforward to analyse $R(T)$ data at fixed magnetic fields to see if there is any feature at H_{c2_A} . For the discussion of their experimental results (and criteria), it would also help if we authors present more details on the sample characterization in the supplement, and e.g. discuss (i) the evolution of residual resistivity ratio as a function of x , (ii) the change of superconducting transition width with x in zero field, and (iii) the phase diagram as a function of x for their samples studied (These might be contained in some of the previous works of the group, but it would be useful to have at hand in the present paper as well).

Small remark: In Fig. 3 of the main text commas in D and H should be replaced by dots.

Dear Editor,

Thank you for sending over the Referees' reports and for offering us the possibility to resubmit the manuscript. We appreciate the detailed and constructive comments from our three referees. We apologize for the delay in responding to their reports but we have followed your editorial recommendation and made substantial changes to the manuscript, as summarized below:

1) We have carried out a complementary analysis of the irreversibility and upper critical fields on the sister family $\text{FeSe}_{1-x}\text{Te}_x$ and have incorporated a plot of the resulting ratio into the manuscript (new Figure 4) and added Supplementary Note 5 in the Supplementary Information (SI) along with selected plots of the raw data in Supplementary Figure 6. Combining our analysis with that performed on the $\text{FeSe}_{1-x}\text{S}_x$ family has enabled us to exclude many of the candidates proposed as the sole origin of the marked expansion of the low- T vortex liquid regime beyond the nematic quantum critical point (QCP). As a result of this, we have modified the title of the manuscript and included K. Ishida, K. Mukasa, S. Imajo, M. W. Qiu, M. Saito, Y. Uezono, T. Otsuka, T. Watanabe and K. Kindo, who were involved in the material synthesis and high-field measurements, as co-authors.

2) The discussion section has been significantly redrafted to reflect these developments and its focus has now switched to the importance of quantum fluctuations of the superconducting order parameter in regions of the phase diagram where the intrinsic superconductivity is weakened.

3) In response to comments from Referee #2, Figure 1 has been modified and the introduction section redrafted accordingly.

4) In response to comments from Referee #1, additional measurements were performed in order to check for any current dependence in the ratio $H_{irr}(0)/H_{c2}(0)$ and included in the SI as new Supplementary Note 1. Accordingly, we have now also included Y.-T. Hsu as a co-author of the manuscript in recognition of his contribution to these measurements.

Please find below our full response to the Referees' comments and more details of the corresponding revisions that we have made to the manuscript. We hope that this revision addresses most, if not all, of the Referees' concerns. The original comments from the Referee(s) are copied in red, our response is in black, while the action we have taken is summarized in blue. All changes to the actual manuscript and SI have also been highlighted in blue.

We look forward to your editorial decision in due course.

Yours sincerely,

Matija Čulo and Nigel Hussey (on behalf of all co-authors)

We thank the Referee for their thoughtful review of our manuscript and appreciate the points that have been raised. Their clarification has certainly helped to improve it. Below we address each point in turn.

*#1.1 In their present manuscript, Culo, Licciardello et al. discuss the evolution of the two critical magnetic fields H_{irr} and H_{c2} of the normal-to-superconducting transitions in the iron-based superconductor $Fe(Se,S)$. They report a significant separation of H_{irr} and H_{c2} towards $T \rightarrow 0K$ across the nematic quantum phase transition which they interpret as a signature of an enhanced Quantum Vortex Liquid regime. They discuss possible reasons and implications for its formation / stabilization, such as the role of AFM and nematic quantum fluctuations and the proximity of $FeSe$ to a BEC-BCS crossover. However, from their data alone they cannot identify the dominant driving force. As such, I feel that the present results are certainly rather interesting, but I do not see how they will advance the wider field of research. Instead, they quite strongly relate to $Fe(Se,S)$ only. As such, I cannot recommend publication in *Nature Communications* but would suggest a more specialized journal.*

The Referee is correct that based on the original study, we were not able to identify the dominant driving force for the appearance of the expanded vortex liquid regime in $Fe(Se,S)$ at low temperatures. In order to address this issue, we have carried out a complementary study on the sister family $FeSe_{1-x}Te_x$ and have incorporated a plot of the resulting ratio $H_{irr}(0)/H_{c2}(0)$ into the revised manuscript (new Figure 4). Combining our analysis with that performed on the $FeSe_{1-x}S_x$ family has enabled us to exclude many of the candidates proposed as a unique origin of the marked expansion of the low- T vortex liquid regime beyond the nematic QCP. In particular, we have been able to rule out the effect of a putative BCS/BEC crossover, nematic or spin fluctuations as well as any quantum critical fluctuations associated with these states. We believe this combined study now offers robust insights into the origins of such a QVL state – providing as they do a number of key ingredients necessary for its realization – and will thus help to improve our understanding of what is an intriguing yet still poorly understood state of anisotropic type-II superconductors. We believe that this finding will serve as a boost for further experimental and theoretical studies of the quantum vortex liquid state and its relation to the long-range order that competes and/or coexists with superconductivity and thus that our revised manuscript is well suited to the broad readership of *Nature Communications*.

New Figure 4 added to the revised manuscript, detailing the evolution of $H_{irr}(0)/H_{c2}(0)$ in $FeSe_{1-x}Te_x$ along with an accompanying discussion (in the Discussion section) ruling out a number of unique candidates for the expansion of the QVL state and identifying the importance of quantum fluctuations of the SC order parameter in regions of the phase diagram where the intrinsic superconductivity is weakened. A new Supplementary Note (5) and Figure (6) have also been added to the SI, while several new co-authors have been added to the revised author list.

#1.2 A broadening of the superconducting transition width in the resistivity for $H \parallel c$ was previously reported across the nematic QCP under pressure (PRL 127 246402 (21)). In fact, the derivatives in Fig.2 (present manuscript) and Fig. 4 (reference above) appear very similar, suggesting they describe the same effect. In contrast to the present paper, the authors above related the broadening directly to the nematic QCP since the transitions become sharp again towards higher pressures. Moreover, the authors above interpreted the temperature dependence of the resistivity between H_{irr} and H_{c2} as evidence for a vortex liquid which would freeze to a vortex glass at zero temperature. Since both reports cover the same temperature and field range, this suggests to rule out a QVL in this regime, plus the pressure vs chemical composition dependence seem to differ outside the nematic phase. Could the present authors comment how their results compare to the reference above?

We thank the Referee for pointing us to the pressure study of Reiss *et al.* Indeed, the derivatives in Fig. 2 of our manuscript look similar to the derivatives in Fig. 4 of their paper, though the former refers to $H \parallel ab$, while the latter refers to $H \parallel c$. Moreover, the evolution of the width of the derivative peaks across the nematic QCP in the two studies is markedly different. While the width is maximized close to the nematic QCP at the critical pressure p_c in Reiss *et al.*, in our study the width is maximized far beyond the QCP. Such behavior appears to rule out the nematic QCP as the main driving force for the enhanced VL regime, as argued on page 8, lines 204-208 in the original manuscript. Note too that a broadening of the VL regime under pressure is also observed beyond the nematic phase [Ayres *et al*, *Commun. Phys.* **5**, 100 (22)]. Lastly, it is interesting to note that the enhancement shown in Reiss *et al.* at the nematic QCP is due to an enhancement in $H_{c2}(0)$, while in our study, the enhancement results from a drop in $H_{irr}(0)$. Hence, contrary to the Referee's suggestion, we believe that the nature and origins of the enhancement in both studies are distinct. This conclusion is confirmed by our complementary study on Fe(Se,Te) which is now included in the revised manuscript. Collectively, these studies provide strong evidence that the enhanced VL regime is in fact linked to a reduction in the strength of superconductivity in both systems.

Regarding the freezing of the vortex liquid (VL) to a vortex glass at zero temperature, the Referee is correct. Indeed, as pointed out also by another Referee, strictly speaking, the VL is absent at 0 K in the 3D case although the field range of vortex liquid is broadened at very low but finite T . Since in our study the VL seems to be present at temperatures as low as $0.03 T_c$, where only quantum fluctuations are expected to be responsible for melting of the vortex glass, we refer to such a regime as a QVL. Nevertheless, in response to the comments of both referees, we have replaced the phase diagram in Fig. 1D with the theoretically correct one and changed the wording accordingly.

Comment on the pressure study of Reiss *et al.* added at the end of the third paragraph in the section Results and reference added to the reference list. Wording throughout the manuscript changed to acknowledge freezing of the QVL to a vortex glass at zero temperature and the phase diagram in Fig. 1D replaced with the theoretically correct one.

#1.3 Current dependence: In the Introduction, the present authors argue that they work in the zero-current limit $j \rightarrow 0$. However, at the lowest temperatures, a previous study e.g. on YBCO demonstrated that H_{irr} may strongly depend on the applied current (PNAS 2021216118) even though for this system, a very large critical current could be assumed. Did the present authors check for a current dependence of their results to support their claim of working in the low-current limit?

We thank the Referee for seeking clarification on this issue. In our original study, we had measured $\rho(H)$ curves at two different currents (0.5 mA and 1 mA) and not noticed any discernible change in $H_{irr}(0)$. However, in response to the Referee's comment, we checked the current dependence of H_{irr} at low T over a much broader range of excitation currents (30 μ A – 1 mA) for the sample FSS25 (for which the QVL regime is the most pronounced). Though the MR curves do slightly differ for different current excitations, H_{irr} changes only within a small range – from 6 T to 7 T – at $T = 0.6$ K with a 30-fold decrease in excitation current. Taking into account the average value of $H_{c2} \approx 20$ T, this implies that the ratio H_{irr}/H_{c2} changes within the range 0.3 - 0.35. The main conclusion of the present study therefore stays unaffected, supporting the approximation of the zero current limit $j \rightarrow 0$.

Figure R1. Current dependence of H_{irr} for FSS25 at $T = 1.8$ K (left panel) and $T = 0.6$ K (right panel).

We have added a new section to the supplement (Supplementary Note 1) and a sentence to the main text at the end of the first paragraph in the Results section summarizing these results. Yu-Te Hsu, who carried out these measurements (and is incidentally the first author of the study highlighted by the Referee), has now been added to the author list.

#1.4 Relation to nematicity / interpretation of the data: The present manuscript is part of an ongoing debate about the role of nematic quantum critical fluctuations on superconductivity and the normal state behavior of Fe(Se,S), as set out in the manuscript at several locations. However, for a precise localization of the nematic QCP, the exact sulfur content of Fe(Se,S) samples was pointed out to be crucial in Ref. [39]. Similar to their previous works (Refs. [35,36,40,41]), the present authors use the nominal sulfur concentrations instead of the EDX values to locate their samples in the phase diagram. Yet, I recall that Ref. [39] claimed that sample FSS20 still showed a nematic transition temperature. In this case, the drop in H_{irr}/H_{c2} would occur well inside the nematic phase, and not across its boundary, and hence it could be related to strong AFM fluctuations. Note for example that Xiang et al. reported the emergence of a magnetic phase inside(!) the nematic phase (PRB 96 024511). Could the present authors clarify the exact location of the nematic phase boundary based on the presently used samples and/or the values of T_s and/or the measured x_{EDX} ? The chemical vapor transport-grown samples are known to show a large scattering of real sulfur contents due to the thermal gradients and such, a per-sample assessment is required.

While EDX measurements on Fe(Se,S) are informative, as the Referee points out, they often show a large variation in the S content both within an individual crystal (typically ± 0.01) and within a single

batch of crystals (± 0.02 , as shown for example in the Xiang *et al.* study). The key point for this particular study is the location of the nematic phase boundary relative to the expansion of the QVL regime. In order to confirm this relation, we compare in the figure below the T -dependence of the Hall coefficient for several S-substituted samples across the doping range $0 \leq x \leq 0.25$. The top panels show $R_H(T, x)$ for the samples used in the present article, while the bottom panels show $R_H(T, x)$ for samples reported in Huang *et al.* [*Phys. Rev. Res.* **2** 033367 (20)] for which the S concentrations were determined using EDX.

In pure FeSe, we find that with decreasing temperature, $R_H(T)$ passes through a positive maximum at $T_{\max} \approx 75$ K before becoming negative at a lower temperature $T_0 \approx 60$ K. With increasing x , both T_{\max} and T_0 shift to lower temperatures until eventually, the crossover to a negative R_H at low T vanishes. In our samples, the disappearance of the crossover in $R_H(T)$ occurs between FSS13 and FSS16 while in the Huang *et al.* study, it occurs between $x = 0.12$ and $x_c = 0.17$. Tellingly, for FSS16 and FSS18, $T_{\max} \approx 30$ K and 15 K respectively, while for the Huang $x_c = 0.17$ sample, $T_{\max} \approx 20$ K, i.e. intermediate between these two values. This suggests that our FSS16 and FSS18 samples are indeed located on opposite sides of the nematic QCP. Moreover, for $x > x_c$, the maximum in $R_H(T)$ is seen to diminish monotonically with increasing x as the system is tuned away from the nematic QCP. The same trend is observed in our most highly doped samples. Overall, we believe this comparison with EDX-determined samples provides strong evidence that the S contents of all samples are within the range of uncertainty for EDX-determined samples and that expansion in the VL regime revealed in our study does indeed set in beyond x_c . Lastly, we could not see any claim in Ref. [39] that FSS20 showed a nematic transition.

Figure R2. Comparison of $R_H(T)$ for $\text{FeSe}_{1-x}\text{S}_x$ as reported in *PRR* **3** 023069 (21) (top panel) and *PRR* **2** 033367 (20) (bottom panel).

In light of the arguments above, no changes to the manuscript were deemed to be necessary in response to this comment.

We are very thankful to the Referee for their thoughtful review, for clarification of certain theoretical issues and for recommending the publication of our manuscript once a number of points had been addressed. Here we address these points.

#2.1 The authors' interpretation on the origin of the broad QVL regime is inconsistent with other published works on the same family of materials, e.g., Ref. 43. For instance, the phrases "enhanced SC fluctuation, not related to vortex motion" on page 3 (line 87) and page 8 (line 223) and "the superconducting quantum fluctuations are too weak" in the abstract are incomprehensible, because strong thermal fluctuations usually enhance the corresponding quantum fluctuations at finite temperatures. The present version of this manuscript cannot be accepted for publication, and a drastic revision of the manuscript is necessary to make the content of the research work readable.

We thank the Referee for raising this point. The statements 'not related to vortex motion' and 'the superconducting quantum fluctuations are too weak' have now been removed. More importantly, by carrying out a complementary study of the vortex liquid regime in the sister Fe(Se,Te) family, we are now able to show that the expansion of the vortex liquid regime at low T is indeed due to the superconducting (SC) state becoming less robust and as a result, SC fluctuations becoming more prominent. This complementary study has also enabled us to rule out the other candidates as the sole origin for this effect.

Statements 'not related to vortex motion' and 'the superconducting quantum fluctuations are too weak' removed and a revised discussion of the dominant role of SC fluctuations due to a weakening superconductivity added to the Discussion section. Alignment with conclusions of Ref. [43] also noted.

#2.2 Obviously, the crucial error in their interpretation on the resistivity data consists in their estimation of the dimensionless quantum resistance Q_u in sec. 3. There, they assume that the friction (damping) coefficient η (in the notation by Refs. 1, 5, and 6) takes the conventional BCS form proportional to the normal conductivity $1/\rho_n$. The coefficient η is found as the coefficient of the dissipative time-derivative term of the time-dependent Ginzburg-Landau equation following from the corresponding electronic Hamiltonian describing the superconductor of one's interest. For unconventional superconductors, such theoretical studies are available; PRB 66, 094513 ('02) for high T_c cuprates and PRB 105, 174510 (22) for FeSe ($x=0$) near $H_{c2}(0)$. In the latter, the vortex friction coefficient η is proportional to η_n there (eqs. 19 and 20 there), while eqs. (26) -- (32) are related to this issue in the former. As seen in these works, the friction coefficient is drastically reduced, i.e., the quantum SC fluctuation is enhanced irrespective of the thermal fluctuation, due to electronic details and/or an effect of other (if any) non-SC orders.

Namely, the friction coefficient may be independent of the normal resistivity ρ_n . In relation to this, the author should understand the following things on how to theoretically describe possible symmetry breakings and orders: The QVL is the state created by strongly interacting SC fluctuations,

and, as far as other possible orderings do not occur within the QVL regime, effects of such other orders or fluctuations are incorporated in the corresponding Ginzburg-Landau free energy F_{GL} expressed by the SC order parameter in a form affecting the coefficient of each term of F_{GL} . Namely, in any case, the broader QVL is a consequence of thermal and/or quantum SC fluctuation enhanced by the decrease and disappearance of the nematic order. In fact, the heat capacity data of $FeSe_{1-x}S_x$ in Ref. 43 have clearly shown the SC fluctuation enhanced drastically with increasing x . It might be possible that the nematic or other quantum critical fluctuation near x_c reduces the SC fluctuation effect. In fact, the authors mention in relation to Refs. 19 and 20 that the H_{c2} is enhanced, implying an enhanced SC order. The resulting reduction of the SC fluctuation competes with the effect of enhancing the SC fluctuation due to the disappearance of the nematic order. In relation to the sentence on the line 149 on page 7, it might be possible that H_{irr}/H_{c2} in $x < x_c$ is nearly constant and has not decreased with increasing x as a consequence of this competition.

Following the Referee's comments and inclusion of the Fe(Se,Te) study, the discussion of the quantum resistance and the strength of SC fluctuations has been significantly modified. Indeed, the new study on Fe(Se,Te) reveals that the dominant driving force for the expansion of the mixed state regime is indeed the reduction in the strength of the superconductivity and corresponding enhancement of the SC fluctuation regime and possible reduction in pinning strength. Hence, while the broader QVL regime in Fe(Se,S) beyond x_c may be related in part to the decrease and disappearance of the nematic order, it is clear that this cannot be the sole reason since an even more significant broadening occurs in Fe(Se,Te) inside the nematic phase. The possible cause of the constant value of the ratio H_{irr}/H_{c2} for $x < x_c$ in Fe(Se,S) mentioned by the Referee (i.e. due to a competition of competing effects) is therefore in marked contrast with what is found in the Te-substituted family.

Discussion of the quantum resistance and the strength of SC fluctuations significantly modified and relevant references added.

#2.3 The authors mention that the vortex pinning is enhanced with increasing x . On the other hand, when an electron system becomes dirtier with increasing x , conventionally the strength of the SC fluctuation tends to be enhanced as a consequence of the reduction of the superfluid density. To summarize, the authors need to revise their explanation on the expansion of the QVL regime.

This sentence has now been removed.

Sentence referring to enhanced vortex pinning strength in disordered samples has been removed.

#2.4 It seems that the references to be quoted should be reconsidered. Experiments which have led to Ref. 10 have been performed in relation to the theoretical review paper treating the 2D case, 1) Int. J. Mod. Phys. 10, 601 ('96). In relation to this, I notice that Ref. 11 is absent (The title is incorrect). Further, Ref. 14 has been summarized based on a support via another theoretical paper treating both the 2D and 3D cases, 2) J. Phys. Soc. Jpn. 72, 2930 ('03) (arXiv:cond-mat/0210626) (In particular, see Figs. 3, 10, and 11 there). In these two papers, the resistivity has been examined as a measure of the quantum fluctuation in QVL. In contrast, Refs. 7 and 8 treat the QVL in a 2D model assuming non-dissipative dynamics and thus, are not comparable with any resistivity data.

We have now rephrased the second paragraph on page 2 starting with “While initial theoretical work...” and have replaced references 7 and 8 with the suggested references. We have also corrected the title in Ref. [11]. We thank the Referee for bringing these oversights to our attention.

Second paragraph on page 2 rephrased and suggested references included. Title of Ref. [11] also corrected.

#2.5 When an expert on the vortex matter in superconductors starts reading this manuscript, he would stop reading the manuscript further by looking at Fig.1A and related sentences on pages 1 and 2, because Fig. 1A is completely erroneous. The ordered phase is the unpinned vortex solid (VS). The unpinned VS known as the Abrikosov lattice is not a VL but an anisotropic SC phase. The VL is induced by the SC fluctuation and is one part of the normal phase in which the resistivity is non-vanishing for any direction of the current, while the resistivity in a current parallel to the magnetic field is zero in the unpinned VS. Further, the authors possibly assumed that the pinning-free limit of real VS is the unpinned VS. At least, in the thermodynamic limit, the H_{irr} or the vortex-glass transition field in the pinning-free limit of a real superconductor approaches the vortex lattice melting line $H_m(T)$ and does not decrease down to a value below H_m . The resulting VS is called as Bragg-glass. See PRB 43, 130 (1991), JPSJ 65, 3998 ('96), and PRB 52, 1242 ('95) for details. At least, the Figure A needs to be deleted from Fig. 1 or to be replaced by a theoretically valid one, and the final sentence on page 1 must be deleted.

We thank the Referee for the clarification and for pointing us our loose terminology as well as the helpful references. We have now reworded that part of the introduction and made a clear distinction between H_{irr} – that marks the onset of non-zero resistivity – and the melting line H_m – that marks the onset of a vortex liquid regime. We have also modified Fig. 1A accordingly.

Relevant section of the introduction reworded and Fig. 1A redrafted.

#2.6 Strictly speaking, the figure D of Fig. 1 is also erroneous: In 3D case at zero temperature, the QVL is absent, i.e., $H_{irr} = H_m$, although the field range of VL is broadened at very low but finite temperatures. This theoretical picture follows by directly applying the analysis in the seminal theoretical work, PRB 31, 7124 (85), to the zero temperature case. Therefore, I respect that the phrase “even at temperatures as low as $0.03 T_c$ ” had been written in sec. 4 in the manuscript. If possible, it would be better that the authors also revise Fig. 1D in this sense.

Again we thank the Referee for raising this point. We have now revised Fig. 1D in accordance with the Referee’s suggestion.

Fig. 1D revised in accordance with the Referee suggestion.

#2.7 Through the paragraph below Fig. 1, I feel that in the VL $F_p < F_L$ is assumed. As mentioned above, however, the VL is one part of the normal phase, and thus, $F_p = 0$ on average there. I feel that the authors consider here the pinning strength for a single vortex. However, the pinning strength for a single vortex is ill-defined in the VL. In any case, the paragraph below Fig. 1 has been written based on the old mean field picture that the H_{c2} line is the SC transition line. It is well known at

present that this old picture is erroneous. In fact, based on this old picture, the non-vanishing resistivity in a current parallel to the field in the VL regime cannot be explained (see JPSJ 58, 1906 (89)).

We thank the Referee for the clarification because indeed, we were considering the pinning strength for a single vortex and assumed that in the VL $F_p < F_L$. We have now rewritten the corresponding paragraph without considering the pinning strength for a single vortex. For clarity, we have now made a clear distinction between the irreversibility line H_{irr} – that marks the onset of non-zero resistivity – and the melting line H_m that marks the onset of a vortex liquid regime.

Paragraph below Fig. 1 rewritten without considering the pinning strength for a single vortex. Also a clearer distinction has now been made between the irreversibility line H_{irr} and the melting line H_m .

#2.8 Finally, regarding the details of data, Fig. 3A (i.e., $x = 0$ and $H // ab$ case) seems to be different from that in Ref. [44]. Further, the authors have never discussed about an effect of the paramagnetic pair breaking which was one of the main issues in Refs. [44] and [46]. At least the H_{c2} curves in A, B, and C of Fig. 3 become flat at lower temperatures possibly reflecting a strong paramagnetic pair-breaking. Discussing about a reason why the paramagnetic pair breaking becomes less effective with increasing x would be interesting. For instance, is it due to the fact that the system becomes dirtier with increasing x ?

The plots in Fig. 3A and those in Fig. 2 of Ref. [44] are in fact derived from the same data set and are thus in excellent agreement – any slight discrepancies between the two phase diagrams will be present only as a result of different criteria being used to deduce H_{irr} and H_{c2} . We thank the Referee for raising the issue of paramagnetic pair breaking – it does appear to be the case that the low- T $H_{c2}(T)$ plots are flatter for lower x values, suggesting that the increased disorder upon increasing S substitution may soften the paramagnetic pair breaking effect. However, since this is not the focus of the present study, we have decided not to include any discussion of this point in the revised manuscript.

No change in the manuscript was made in response to this comment.

Again, we thank the referee for their thoughtful review of our manuscript and for recommending its publication once a number of points had been addressed. Here we address these points.

#3.1 The main concern relates to the interpretation of the finding of the expanded QVL state in connection with the role of quantum critical degrees of freedom of magnetic/nematic origin. The authors themselves attest that the “absence of any sharp dip in H_{irr}/H_{c2} near the QCP”, but do not take that into account when making connections to quantum criticality. Obviously, the step-like change that they observe does not meet the intuitive expectations of quantum critical fluctuations being important, but rather points to some discontinuous change of some of the superconducting parameters (e.g. like gap parameters). Therefore, all arguments regarding the importance of magnetic vs. nematic fluctuations to be important appear very speculative.

We agree with the Referee’s assertion that the importance of magnetic/nematic fluctuations for the appearance of the expanded vortex liquid regime in Fe(Se,S) at low T was somewhat speculative in the original manuscript. In order to address this issue (a point also raised by Referee #1), we carried out a complementary study on the sister family FeSe_{1-x}Te_x and have incorporated a plot of the resulting ratio $H_{irr}(0)/H_{c2}(0)$ into the revised manuscript (new Figure 4) along with a new Supplementary Note (5) and Figure (6) in the SI. Combining our analysis with that performed on the FeSe_{1-x}S_x family has enabled us to exclude many of the candidates proposed as a unique origin of the marked expansion of the low- T VL regime beyond the nematic QCP. In particular, we have been able to rule out the effect of a putative BCS/BEC crossover, nematic or spin fluctuations as well as any quantum critical fluctuations associated with these states. Nevertheless, it is likely that their combined presence, coupled with the low dimensionality, creates the necessary conditions for quantum SC fluctuations to have a profound destabilising effect on the vortex solid, one that is amplified wherever the intrinsic superconductivity is weakened. We thus believe that this combined study offers robust insights into the origins of the QVL in unconventional superconductors and will thus help to improve our understanding of what is an intriguing yet still poorly understood state.

New Figure 4 added to revised manuscript, and new Supplementary Note (5) and Figure (6) added to the SI detailing the evolution of $H_{irr}(0)/H_{c2}(0)$ in FeSe_{1-x}Te_x along with an accompanying discussion (in the Discussion section) ruling out many of the candidate origins for the expansion of the QVL state and identifying the importance of SC fluctuations in regions of the phase diagram where the intrinsic superconductivity is weakened. Several co-authors have also been added to the revised author list.

#3.2 The second concern relates to the determination of H_{irr} and H_{c2} from the transport data. Even though the authors take a great effort in the main text and in the supplement to justify that their finding is independent of the choice of criterion, I still feel somewhat uncomfortable when seeing the absence of any feature at H_{c2_A} in Supplementary Figure 4 E and F. In the main text, the authors argue that the longitudinal magnetoresistance is negligible for high x , and therefore MR is straightforward to analyse. If this is the case, then I would also expect that it should be straightforward to analyse $R(T)$ data at fixed magnetic fields to see if there is any feature at H_{c2_A} .

Figure R3. $d\rho/dH$ plots corresponding to the field sweeps shown for FSS13 and FSS25 in Supplementary Fig. 5E and 5F of the revised manuscript. The vertical arrows indicate the location of H_{c2}^A in both plots.

We agree with the Referee that the feature at H_{c2}^A is hard to discern from the MR field sweeps. Indeed, the feature is only visible when plotting the derivative, as shown above for the two sweeps in question. We have now included these plots in Supplementary Fig. 2 so that the reader can compare them directly with the raw $\rho(H)$ curves in Supplementary Fig. 5. Note that at $H = H_{c2}^B$ – the value for H_{c2} deduced by method B in Supplementary Fig. 5 – the slopes of the MR curves are approximately twice their expected value at H_{c2}^A (i.e. in the absence of SC fluctuations), implying that H_{c2}^B values are very much an underestimate. Due to the cost in energy of performing temperature sweeps at constant field at the high-field facility, we did not carry out a comprehensive set of $\rho(T)$ curves at fixed magnetic fields. We hope that the $d\rho/dH$ plots shown below will allay any concerns that the Referee had about the determination of H_{c2}^A .

In response to this comment, we have now included these 1.3 K derivative plots in a revised Supplementary Figure 2 to enable the reader to compare them with the raw $\rho(H)$ curves in Supplementary Figure 5.

#3.3 For the discussion of their experimental results (and criteria), it would also help if we authors present more details on the sample characterization in the supplement, and e.g. discuss (i) the evolution of residual resistivity ratio as a function of x , (ii) the change of superconducting transition width with x in zero field, and (iii) the phase diagram as a function of x for their samples studied (These might be contained in some of the previous works of the group, but it would be useful to have at hand in the present paper as well).

This we have now done.

More details on the sample characterization provided in the SI, as requested by the Referee.

#3.4 Small remark: In Fig. 3 of the main text commas in D and H should be replaced by dots.

We have now replaced the commas in Fig. 3D and 3H with dots.

REVIEWERS' COMMENTS

Reviewer #1 (Remarks to the Author):

In their revised manuscript, Culo, Licciardello et al. have made substantial improvements compared with their original submission, in response to the numerous comments raised by the referees. In particular, the authors have expanded their experimental dataset by studying also the Fe(Se,Te) doping series for which there was a recent advancement in single crystal growth. The new data clearly improve the quality of the manuscript, not only because they extend the significance of the results presented, but also because they allow the authors to strengthen or exclude a number of previous tentative or speculative explanations. As such, I expect that the present study will stimulate further experimental and theoretical work to uncover the origin of the enhanced VL regime. I therefore recommend publication of the present manuscript.

Reviewer #2 (Remarks to the Author):

In the second (revised) manuscript, the authors seems to have convincingly responded to requirements and recommendations given in my first report. In the second manuscript, there is one minor but confusing description on the line 201 of the second manuscript: The $1/\rho_n$ in the Q_u - expression there should be η . Otherwise, readers could not understand the meaning of the ensuing sentences naturally. Except this point, there are no parts to be repaired further in the second manuscript, and I will be able to recommend publication of the present work.

Reviewer #3 (Remarks to the Author):

The authors have taken great care in revising the manuscript, and it is greatly appreciated that they took the effort to also include data on Te-doped FeTe to better narrow down what factors determine the extent of the quantum vortex liquid regime.

However, in line with comments from the other Referees, it still appears that much of the argumentation is a bit speculative. The authors themselves state in their concluding remarks:

"The persistence of this broad QVL regime over such an extended region of the phase diagram, however, suggests that key elements of the electronic state of iron chalcogenides, such as the presence of strong nematic and magnetic fluctuations and proximity to the BCS-BEC crossover, conspire to destabilise the vortex solid to such an extent that any further weakening of the SC order leads to the QVL regime being amplified".

This indicates that there is still - despite all the efforts of including Te-doped FeSe -no detailed knowledge on the driving force available. In addition, the title of the manuscript suggests that the manuscript would establish a direct link of the quantum vortex liquid to nematicity, which is not the case.

At this point, even after the revisions, I would agree with Referee #1 that the manuscript is better suited in a more specialized (Nature) journal and would hope that the presented results indeed stimulate further theoretical work into the origins of extended quantum vortex liquid states.

Dear Editor,

Thank you for sending over the Referees' reports and for offering us the possibility to publish our manuscript, after minor changes have been made. Please find below our response to the Referees' comments and the corresponding changes that we have made to the manuscript. As done previously, the original Referee comments are in red, our response is in black, while the action we have taken is summarized in blue. The changes in the manuscript have also been highlighted in blue.

Please note that the new version of the manuscript has been prepared in accordance with the editorial requirements listed in the document provided in your last email. The footnotes have been removed from the reference list and integrated into the main text, which is also highlighted in blue. In addition, the figure panels are now labelled with lower-case, boldface 'a', 'b', etc. placed in the top left corner and the references to them are adjust accordingly throughout the manuscript.

Yours sincerely,

Matija Čulo and Nigel Hussey (on behalf of all co-authors)

Report of the First Referee – NCOMMS-22-15468/Culo

#1.1 In their revised manuscript, Culo, Licciardello et al. have made substantial improvements compared with their original submission, in response to the numerous comments raised by the referees. In particular, the authors have expanded their experimental dataset by studying also the Fe(Se,Te) doping series for which there was a recent advancement in single crystal growth. The new data clearly improve the quality of the manuscript, not only because they extend the significance of the results presented, but also because they allow the authors to strengthen or exclude a number of previous tentative or speculative explanations. As such, I expect that the present study will stimulate further experimental and theoretical work to uncover the origin of the enhanced VL regime. I therefore recommend publication of the present manuscript

We thank the Referee for their second review of our manuscript and for recommending its publication in *Nature Communications* in the present form.

No changes to the manuscript have been made in response to this comment.

Report of the Second Referee -- NCOMMS-22-15468/Culo

#2.1 In the second (revised) manuscript, the authors seems to have convincingly responded to requirements and recommendations given in my first report. In the second manuscript, there is one minor but confusing description on the line 201 of the second manuscript: The $1/p_n$ in the Qu -

expression there should be η . Otherwise, readers could not understand the meaning of the ensuing sentences naturally. Except this point, there are no parts to be repaired further in the second manuscript, and I will be able to recommend publication of the present work.

We thank the Referee for their second review of our manuscript and for recommending its publication in *Nature Communications* after this minor change has been made. We have now replaced $1/\rho_n$ in the Q_u expression with η and adjust the corresponding sentence on line 209 in the new version of the manuscript.

The term $1/\rho_n$ replaced by η in the Q_u expression and the corresponding sentence adjusted accordingly on line 209 in the new version of the manuscript.

Report of the Third Referee -- NCOMMS-22-15468/Culo

#3.1 The authors have taken great care in revising the manuscript, and it is greatly appreciated that they took the effort to also include data on Te-doped FeTe to better narrow down what factors determine the extent of the quantum vortex liquid regime. However, in line with comments from the other Referees, it still appears that much of the argumentation is a bit speculative. The authors themselves state in their concluding remarks: "The persistence of this broad QVL regime over such an extended region of the phase diagram, however, suggests that key elements of the electronic state of iron chalcogenides, such as the presence of strong nematic and magnetic fluctuations and proximity to the BCS-BEC crossover, conspire to destabilise the vortex solid to such an extent that any further weakening of the SC order leads to the QVL regime being amplified". This indicates that there is still - despite all the efforts of including Te-doped FeSe - no detailed knowledge on the driving force available. In addition, the title of the manuscript suggests that the manuscript would establish a direct link of the quantum vortex liquid to nematicity, which is not the case. At this point, even after the revisions, I would agree with Referee #1 that the manuscript is better suited in a more specialized (Nature) journal and would hope that the presented results indeed stimulate further theoretical work into the origins of extended quantum vortex liquid states.

We thank the Referee for their second review of our manuscript and for recognizing its importance and potential to stimulate further work on the issue of quantum vortex liquid. We respectfully disagree, however, with the Referee's assertion that, even after including the data for FeSe_{1-x}Te_x, we are still unable to determine the main driving force behind the broad QVL regime observed in this study. Rather, by including the comparative study on FeSe_{1-x}Te_x, we have been able to uncover a striking correlation between the extent of the QVL and the strength of superconductivity, which clearly gives new insights into how the QVL regime itself can be enhanced. Given that weak-coupling superconductivity (i.e. with a small gap amplitude) cannot realize a broad QVL regime on its own, it is clear to us that the strong nematic and magnetic fluctuations, as well as proximity to the BCS-BEC crossover, present in these chalcogenide superconductors are essential ingredients to enable the vortex solid to become destabilised at low temperatures.

With regards to the title, we would like to clarify that while the title of the original manuscript implied a direct link between nematicity and the quantum vortex liquid, the title has since been changed to avoid such a direct link. Rather, the new title reflects simply the fact that these expanded QVL regimes are observed in two superconducting systems that are, for a large part of their respective phase diagrams, also electron nematics. We are confident that the new wording of the title does not lead to ambiguity in the mind of the reader and hope that the Referee can concur with this.

No changes to the manuscript have been made in response to this comment.